



*Climate Stories*: Enabling and sustaining arts interventions in climate science communication

Ewan Woodley[1], Stewart Barr[1], Peter Stott[2,3], Pierrette Thomet[4], Sally Flint[5], Fiona Lovell[4], Evelyn O'Malley[6], Dan Plews[4], Chris Rapley[7], Celia Robbins[1], Rebecca Pearce[1], Rebecca Sandover[1]

[1]Geography, College of Life and Environmental Sciences, University of Exeter, UK
[2]Mathematics, College of Engineering, Mathematics and Physical Sciences, University of Exeter, United Kingdom
[3]Hadley Centre, Met Office, Exeter, United Kingdom
[4]Independent artist
[5]English, College of Humanities, University of Exeter, United Kingdom
[6]Drama, College of Humanities, University of Exeter, United Kingdom
[7]Earth Sciences, University College London, United Kingdom

Correspondence to: Ewan Woodley (E.J.Woodley@exeter.ac.uk)

**Abstract**

The climate science community faces a major challenge for communicating the risks associated with climate change within a heavily politicised landscape, characterised by varying degrees of denial, scepticism, distrust in scientific enterprise and an increased prevalence of misinformation ("fake news"). This issue is particularly significant given the reliance on conventional 'deficit' communication

approaches, which are based on the assumption that scientific information provision will necessarily lead to desired behavioural changes. Indeed, we argue that the constrained orthodoxy of scientific practices in seeking to maintain strict objectivity and political separation imposes very tangible limits on the potential effectiveness of climate scientists for communicating risk in many contemporary settings. To address these challenges, this paper uses insights from a collaboration between UK climate

scientists and artist researchers to advocate for a more creative and emotionally attentive approach to climate science engagement and advocacy. In so doing, the paper highlights innovative ways in which climate change communication can be re-imagined through different art forms to enable complex concepts to become knowable, accessible and engaging to wider publics. We demonstrate that in learning to express their work through forms of art, including print-making, theatre and performance,

song-writing and creative writing, researchers experienced not only a sense of liberation from the rigid





communicative framework operating in their familiar scientific environment, but also a growing self-confidence in their ability and willingness to engage in new ways of expressing their work. As such, we argue that scientific institutions and funding bodies should recognise the potential value of climate scientists engaging in advocacy through art-science collaborations and that these personal investments

and contributions to science engagement by individuals should be rewarded and valued alongside conventional scientific outputs.

## 1. Introduction

Recent advances in climate science have led to a scientific consensus recognising the influence of anthropogenic activities on climate change (IPCC, 2018, Oreskes, 2018). However, widespread and

sustained action to tackle anthropogenic climate change ('climate change', hereafter) has not materialised and current actions (frequently framed as behavioural changes) are inadequate to avoid the worst climate trajectories and impacts (Wong-Parodi and Feygina, 2020). We suggest that an important part of this disconnect relates to the entrenched practices prevalent in science communication strategies and in this paper, we argue that dominant deficit approaches to communicating climate change are

unlikely to be effective in engaging publics because they make an implicit assumption that knowledge production and dissemination provide the necessary conditions for engagement and 'rational' decision-making by publics on environmental risks (Cook and Zurita, 2019). In practice, conventional approaches to communicating climate change not only create an epistemic distance between scientists and their intended audiences (Barr and Woodley, 2019), they often fail to generate inspiration and

connectivity through presenting science-heavy material (Roosen et al., 2018) rather than a sustained, deep and emotional engagement between scientists and publics. Furthermore, trust in the scientific enterprise itself has been eroded through recent shifts in science-society relations. The conditions necessary for distrust in climate science stem from the highly politicised nature of climate change (Lee et al., 2018), and from recent transformations in the nature of climate change debates with which publics

engage. Take, for example, the increasing prevalence of 'fake news', and the associated emergence of post-truth politics, both of which have challenged the scientific community to find effective responses to maintain its status and trust among publics (Cook et al., 2018; Lazer et al., 2018). Consequently, we





argue that in order to enhance the potential effectiveness of climate science communication, it is necessary for scientists to reflect critically on these recent developments and to be prepared to radically

adapt their communications strategies to engage meaningfully with their audiences.

To achieve this, we argue for increased emphasis on science advocacy (which we define here as defending and promoting the credibility and value of scientific research) within the climate science community to better enable climate researchers to both appreciate and navigate the societal context in which science operates. This can only be achieved if scientific institutions both value and support these

activities through professional training, communities of support and career recognition. Nonetheless, we suggest that existing conceptualisations of science and advocacy in a binary or categorical manner, or on a science-advocacy continuum, may have limited value for climate scientists seeking to engage in more radical forms of climate communication and engagement.

In addition, engagement in advocacy alone does not address the deficiencies of the deficit model within

scientific practice. As such, we explore ways in which climate change may be made more emotionally connected and engaging to a diverse range of publics, as opposed to conventional models of communication which rely heavily on outdated modes of presenting scientific information (Cook and Zurita, 2019). To do this, we present findings from a research project in which climate scientists collaborated with artist researchers to explore how climate change can be conceptualised and

represented through different forms of art (Rödder, 2017; Burke et al., 2018). This paper focuses on the opportunities that art-science collaborations present to climate scientists for exploring new ways of engaging in climate science communication and engagement with publics. In so doing, we argue that these learning opportunities can present climate scientists with important opportunities to break with the epistemological constraints of scientific practice by engaging with new ways of seeing,

understanding and telling (their own) stories about climate change (Galafassi et al., 2018). In addition, we argue that art-science collaborations offer spaces of possibility for enabling climate scientists to establish the knowledge and confidence required to sustain arts-based interventions in their own science communications for engaging publics with climate change.



The paper is structured in the following way. First, we consider the challenges that face the climate

science community in communicating environmental risks. In particular, we demonstrate how recent

shifts in science-society relations have not been met by necessary changes in the way in which science

is communicated to publics. Second, we demonstrate how existing conceptualisations of science-

advocacy are dominated by a scientific framing, offering little utility to scientists seeking to expand

their interests into more radical forms of climate communication and engagement. Third, we explore

the potential that exists for engaging climate scientists with new ways of seeing and understanding

climate change through disciplines in the arts. In particular, we illustrate how the process of art-science

collaboration may be capable of transforming both the outlook of climate scientists towards science

communication, as well as providing the foundation for sustained interventions in scientific practice.

Using an empirical example from the UK, we demonstrate that engaging in art-science collaborations

offer climate scientists opportunities for gaining increased personal and professional confidence,

enhanced and widened intellectual engagement with climate change, as well as opportunities for

creating new and potentially effective means of engaging publics with climate change and its impacts.

## 2.   Recent challenges to science communication

The scope, complexity and uncertainty of climate change make it a challenging subject to communicate

to non-specialists (Pidgeon and Fischhoff, 2011). Furthermore, the causes of climate change are

invisible and the impacts are seen by many to be both temporally and geographically distant (Moser,

2010). Whilst these challenges alone are significant, further difficulties arise from individuals and lobby

groups who reject the scientific consensus on climate change; instead using a range of strategies in

public and political arenas to oppose measures for climate mitigation or adaptation (Farmer and Cook,

2013; Fischer, 2019). Over the past two decades, these challenges have led to a significant expansion

of research within the social sciences aimed at improving understanding of the climate communication

process (Ballantyne, 2016; Moser, 2016; Fischhoff, 2019). Drawing heavily on cognitive and social

psychology, research has explored a wide range of challenges, from seeking to understand attitudes to

risk, mental barriers, and strategies for inducing behaviour change, to the ways in which climate



scientists interact with a range of audiences (policymakers, the media, stakeholders) (Nerlich et al.,
2010).

Scientific institutions are faced with the continual challenge of explaining and justifying their work, not
only to policymakers but also to society as a whole (Myers et al., 2017). To this end, efforts to
communicate climate science have largely followed a 'knowledge-deficit' perspective in which
'deficient' knowledge among non-specialist individuals is assumed to be the cause of divergent
opinions between scientists and publics (Nabi et al., 2018). Indeed, this approach has formed the basis
for extensive programmes of climate outreach and engagement in the United Kingdom, the United
States and Australia (Corner and Groves, 2014). However, a significant body of psychological research
has demonstrated that the knowledge-deficit model fails in practice as individuals tend towards
dismissal or selective interpretation of scientific evidence in situations where it contrasts with their own
ethics, values or world views (Groves, 2019). In addition, sociologists have long recognised the limited
utility and potentially counterproductive nature of deficit approaches to science communication
(Wynne, 1993). For example, Bauer et al. (2007: 84) assert that:

 "The deficit model is a self-serving rhetorical device and at the heart of a vicious circle: a
deficient public cannot be trusted. Mistrust on the part of scientific actors is returned in kind by
the public".

Yet, despite early recognition of these substantial flaws in the deficit model, and continued criticism
since, there remains a widespread reliance on this approach for climate science communication (Rapley,
2012), which is often illustrative of the substantial disconnect between the climate science community
and the complexity and diversity of the attitudes and behaviours of publics (Woodley, 2019). Crucially,
the deficit model remains the foundation for how many climate scientists both imagine and conduct
their interactions with publics (Cook and Overpeck, 2019).

To compound these issues of communication, there are growing pressures on the interface between
science and society that raise the question of trust in the scientific enterprise itself (Hopf et al., 2019).
Whilst scholarly disagreement exists on how 'trust' should be conceptualised, there is a general



acceptance that it relates to "a psychological state comprising the intention to accept vulnerability based upon positive expectations of the intentions or behaviour of another" (Rousseau et al., 1998; cited in Myers et al., 2017: 845). As such, scientific organisations and climate scientists are acutely aware of the importance of maintaining trust by publics as a means of sharing their specialist knowledge

(Goodwin and Dahlstrom, 2014; Sarathchandra and Haltinner, 2020). At one level, this challenge is not new as climate science has invoked knowledge controversies and partisanal standpoints since its emergence on the political agenda in the mid-late 1980s. Indeed, a significant body of research has demonstrated that climate scepticism and climate denial may be associated with particular demographic variables, as well as with political persuasion, values and worldviews (Hornsey et al., 2016;

Sarathchandra and Haltinner, 2020). Yet crucially, recent changes in media landscapes, alongside increasingly polarised political environments, have endangered the value of science as a whole. Technological developments in media have facilitated the circulation of 'fake' news, misinformation and disinformation, leading to distrust in both the scientific enterprise and misperceptions of scientific knowledge (Iyengar and Massey, 2019). Although 'fake news' is not a new phenomenon, its potentially

deleterious influence has been intensified through widespread use of social media platforms (Lutzke et al., 2019), causing the scale of this threat to scientific credibility to become a focus of recent scientific debate (Scheufele and Krause, 2019). Importantly, these new modes of reaching publics have enabled any individual or group to publish material related to the climate change issue in a globalised, instantaneous, and widely accessible manner, regardless of the veracity of their contributions. Through

these 'post-truth' developments in which deception is commonplace, statements are able to make implicit or explicit appeals to emotion, as opposed to criteria that permit them to be checked effectively (Groves, 2019). As such:

> "…populist campaigns that have acquired wide currency in the last few years have been ontologically predicated on the idea that there exists different 'truths'" (Prasad, 2019: 1217).

In broad terms, these efforts by vested interest groups have not only cast doubt on the scientific consensus on climate change, they have also strengthened existing political polarisation and have constrained societal engagement with this issue (van der Linden et al., 2017). This has facilitated an



erosion in trust by publics in science as a key form of knowledge (Mann & Brevini, 2017: Engels,

2019). Consequently, we argue that scientists must explore and adopt novel modes of engaging with

publics that allow for a deeper connection to the issues raised through climate change research.

### 3. Frameworks for understanding climate change advocacy

Recent challenges to climate science communication have stimulated intense debate within the science

community over how to respond effectively to the transformed cultural context in which science

operates (Groves, 2019). Accordingly, some specialists have become prolific climate science

communicators, most often using online platforms to share research, defend scientific findings, and

discuss climate change with a heterogeneous range of actors (Walter et al., 2019). Indeed, there appear

to be many motivating factors behind those who engage in climate science communication, from a

'strong sense of duty', to opportunities for career advancement (Nisbet and Markowitz, 2015; Sharman

and Howarth, 2017: 835). Conversely, many climate scientists may not engage in climate science

debates, particularly online, through fear of misinterpretation or exploitation of communications (Post,

2016; Entradas et al., 2019). Alongside this, scientists may be wary of engagement due to the existing

pressures of work (Boykoff and Oonk, 2018), through fear of promoting jealously among colleagues,

jeopardising career development, negatively impacting perceptions of science (Rapley and De Meyer,

2014), or through fear of misrepresenting science within the academic community.

Central to this communication debate is the challenge of how scientists address the balance between

what they perceive as science (being honest), and what they perceive as advocacy (being effective)

(Schmidt, 2015). Early research presented this challenge as a "double ethical bind" in which a tension

exists between a loyalty to the scientific method and associated limits to knowledge, and a desire to

raise awareness of the risks that climate change poses to society (Schneider, 1988: 113). In practice,

this framing suggests that a scientist becomes an advocate when a subjective judgement is made

regarding actions that society 'should' take, as opposed to an objective scientific statement based on

evidence (Donner, 2014). Importantly, this dichotomous conceptualisation posits a neutral scientific

endeavour against acts of advocacy, and in so doing, masks the complexities of both scientific practice

and the behaviour of individual scientists. Take, for example, the authority of scientific practice that





stems from scientists following a disinterested and objective approach to the generation of knowledge

(Corner and Groves, 2014). Despite calls from policy makers and the media for neutral scientific

assessments (Safford et al., 2020), it is widely acknowledged that science cannot be regarded as entirely

value-free since research perspectives, framings and practices are often influenced by personal and

institutional experiences (Tadaki et al., 2015). In addition, the values of scientists present themselves in

routine academic activities, such as applications for funding, scholarly presentations and review of

research articles (Donner, 2014). Crucially, whilst these occurrences do not impact upon the validity or

importance of climate science outputs, they highlight that any conceptualisations of science and

advocacy in a binary or categorical manner (Lackey, 2007; Pielke, 2007; Rapley and De Meyer, 2014)

do not reflect the realities of scientific practice.

In seeking to address the simplicity of categorical approaches to defining advocacy, Donner (2014)

proposes a science–advocacy continuum in which a researcher can use research and critical self-analysis

to adopt a scientific approach to understanding advocacy. In this way, the relative contribution of

objective (science-dominated) and subjective (advocacy-dominated) judgements in communications

may be explored to enable a researcher to choose an appropriate place along a continuum. Whilst this

conceptualisation has not overcome scholarly disagreement on the definition of advocacy (Kotcher et

al., 2017), its contribution is important in two ways. Firstly, the 'traditional' binary view adopted by

many climate scientists leads to communications that commonly seek to 'stick to the science'; however,

this approach fails to acknowledge that to some degree, all statements represent advocacy through the

influence of normative judgements (Donner, 2017). Secondly, although scientists are likely to consider

the impact of findings on both journalists and public debate (Post, 2016), it is the audience that cast

judgement on whether they believe a scientist is implicitly advocating for a particular cause. Therefore,

in order to improve engagement with climate science communication, the climate science community

needs to develop a greater understanding and appreciation of the ways in which their own knowledge,

motivation and cultural values impact upon their statements (Donner, 2017). Moreover, it has been

argued that climate science communications and engagements with publics should not only set out the

values held by scientists, but also clearly establish what scientists are advocating for. In this way, a



communication may advocate for more informed public understanding or debate, greater research funding, or a specific policy position (Schmidt, 2015).

In practice, this requires scientists to make the often difficult decision of where on a science-advocacy continuum they feel comfortable based on their personal values and those of the organisations that they represent. Beall *et al*. (2017) suggest that this is necessary because science advocacy has the potential to directly impact perceptions of scientific credibility, as well as the perceived motives of individual scientists. Yet, whilst the science-advocacy continuum (Donner, 2014) may be of value for mainstream communications, we argue that it is of limited utility to climate scientists who wish to explore more radical and experimental ways of engaging people with climate science through different art forms. Firstly, whilst designed as a supportive tool for researchers, the science-advocacy continuum positions the field of communication within a wholly scientific framework, and as such, may serve to constrain the ambitions of scientists to a set of established and recognised approaches to knowledge dissemination and outreach, acting as a yardstick for professional practice. Secondly, the continuum implies that it is both possible and desirable for a researcher to locate themselves between science and advocacy. However, radical means of engaging people with climate change often seek to mobilise science to engender curiosity and initiate interpretation and debate, without, for example, a piece of art carrying explicit reference to a specific advocacy position. Thirdly, the use of the continuum does not appreciate the multiplicity of communication and engagement styles that may be adopted by an individual climate researcher. For example, it is possible for an individual to participate in established forms of science communication whilst also engaging in creative artistic practices to mobilise their research and experiences in an attempt to foster wider non-academic engagement. As such, we agree that understandings of the concept of advocacy are essential for climate scientists (Donner, 2017; Schmidt and Donner, 2017); however, we argue that attempts to accurately define and adopt an advocacy position (for example, along the science-advocacy continuum) places a restrictive and unrealistic burden on researchers seeking to use radical arts-based practices for science communication and engagement.

### 4. Emerging climate change conversations through the arts



Most policy efforts to communicate climate science have sought to bring about cognitive engagement

with publics through the provision of scientific information and rational arguments (Burke et al., 2018). However, the one-way (deficit) model of science communication is hindered by an inability to address the ways in which people perceive and react to information on climate change as an issue (Illingworth et al., 2018). In the broadest sense, the delivery of abstract science-based information not only fails to inspire people, it also lacks the dimension of storytelling required to make information both accessible

and engaging (Roosen et al., 2018). Alongside this problem, the common perception of climate change as a geographically and temporally distant threat presents additional barriers to creating vivid, personally relevant and affective images of climate change in the minds of publics (Nurmis, 2016). As a result, these challenges have led to increased artistic engagement with climate change which, over the past decade, has principally been framed as an accessible means of connecting people with phenomena

that are both unpredictable and difficult to comprehend (Galafassi et al., 2018).

Collaboration between artist researchers and scientists is not a new occurrence (Brown et al., 2017). Since "The Two Cultures" lecture in 1959 (Snow, 2013), scholars have argued that greater cooperation between art and science may be capable of fostering transformative social change (Honeybun-Arnolda and Obermeister, 2019). Yet, the recent surge of interest by artist researchers in climate change has

been borne out of new cultural-political factors, including a recognition of the significance of climate change as a societal problem, and of the deficiencies of established modes of science communication (Sleigh and Craske, 2017; Roosen et al., 2018). Arguably, the key challenge for those engaging in arts-science collaborations is that of using image and narrative to successfully engage publics with chronic hazards such as climate change that are "slow-moving and long in the making" (Nixon, 2011:3; Nurmis,

2016). In this respect, the arts may provide ways of addressing the 'affective gap' through reaching diverse audiences that are not open to traditional methods of science communication (Burke et al., 2018). For example, creative practices in the arts and humanities allow climate change to be expressed through new forms of representation and emotive experiences (Aragon et al., 2019). In so doing, art has the capacity to encourage independent thought and engagement with climate-related issues in a personal

and immediate manner (Capstick et al., 2018). As such, art may be seen as:



"…a process of opening up imaginative spaces where audiences can move freely and reconsider the role of humans as responsible beings with personal agency and stakes in a changing world" (Galafassi et al., 2018: 77).

Nonetheless, of equal importance to the 'result' of art-science collaborations, are the nature of the

collaborations themselves. Artist researchers have enabled scientists to permeate cultural spaces in order to facilitate discourses on climate science with publics (Buckland, 2012). Indeed, scientists have reported gains in personal and professional confidence, including a reconnection with a creative dimension that was professionally suppressed through adherence to scientific protocols and conventions (Glinkowski and Bamford, 2009). Yet, despite the many potential benefits, artist researchers have noted

that such collaborations run the risk of revealing power relations, which most commonly manifest in a uni-directional way in which science has the upper hand (Sleigh and Craske, 2017). Crucially, successful collaborations must move beyond any notion of the arts and humanities merely as a vehicle for translating scientific knowledge into meaningful art (Hulme, 2011). To achieve this, those involved must grapple with the significant task of critically exploring and breaking down the knowledge

hierarchies and disciplinary siloes that both scientist and artist researchers inhabit in their everyday practices. This necessitates artist researchers and scientists developing often uncomfortable discourses in an attempt to shift their ontological and epistemological presumptions (Brown et al., 2017). Accordingly, this task calls for a reflection on whether the primary value of collaboration lies more in the process, rather than the end product (Webster, 2006; Rodder, 2017).

In addressing the challenges inherent in art-science collaboration, it is clear that both the social sciences and humanities must be more strongly integrated with climate science research. Primarily, this call stems from the growing recognition that traditional dichotomous framings, such as those between fact and value, are of limited use in promoting understanding or engagement with contemporary environmental challenges (Galafassi et al., 2018). Alongside this, the way science is intellectually

positioned within Higher Education needs to be evaluated. For example, the distance between science and arts disciplines must be narrowed, as STEM subjects alone are unable to tackle a problem such as





climate change (Hulme, 2011). Moreover, there is a need to create pedagogic interruptions in science to:

> "…place us in new relations with what we already 'know' or, more importantly, that which we
> 305    do not yet and we cannot yet know" (Higgins et al., 2019: 160).

Finally, we argue that climate scientists should seek to further explore the role and importance of narrative in their communications (Howarth et al., 2020). In particular, those working in the humanities are well placed to engage with scientists to explore the potential for developing climate stories as a more engaging means of starting climate change conversations with diverse audiences (Hulme, 2011).

**5.  Methodology**

The research underpinning this paper is motivated by a desire to understand the challenges that pervade climate science communication as set out previously. As such, we detail how an art-science collaboration set out to explore the ways in which climate scientists can engage with different art forms to develop novel and more effective ways of engaging publics with climate change. The research project

(*Climate Stories*) built on the UK's national WAMfest (Weather, Arts and Music Festival), a series of explorations of weather and climate through song recitals, theatre and performance, talks and festivals. Indeed, these WAMfest events highlighted not only the problems inherent with traditional modes of science communication, but also the popularity and potential for mobilising the arts to provide more engaging narratives of climate change. Subsequently, the *Climate Stories* project was funded as part of

the Natural Environment Research Council (NERC) Engaging Environments Programme.

*Climate Stories* set out to establish an environment that encouraged scientists to learn new (non-scientific) ways to see and understand climate change, as well as one that was conducive to critical self-reflection on the practice of science communication. To achieve this, a collaborative methodology was adopted whereby active engagement and interaction among participants formed the basis for working

towards a common goal (Nokes-Malach et al., 2015). Through this approach, *Climate Stories* aimed to foster intense social learning (including in a residential context) among climate scientists to explore innovative ways of communicating climate change to publics. Importantly, for social learning to be





achieved, a change in understanding must not only occur within individual participants, but also more widely within a community of practice (Reed et al., 2010). Therefore, the project sought to explore the

extent to which effective art-science collaboration was able to create climate art and, in the process, create sustained interventions in the way that participating scientists engaged in science communication.

Nineteen participants took part *Climate Stories* and these individuals comprised climate scientists from the Met Office and the University of Exeter who responded to an open call for expressions of interest in the project. Participants ranged from postgraduate students to senior climate scientists, although the

majority of those taking part were at an early stage in their career. In addition, experienced arts practitioners developed the key learning concepts of the project and were responsible for coordinating workshops on printing making, creative writing, theatre and performance, and song-writing, which made up the key structured learning opportunities for participants.

**Table 1**

**Full list of participants in the Climate Stories project**

| Participant identifier | Contextual information |
|---|---|
| HL | University of Exeter |
| FB | University of Exeter |
| GT | University of Exeter |
| DS | University of Exeter |
| LM | University of Exeter |
| WP | University of Exeter |
| CF | University of Exeter |




| OB | Met Office |
|---|---|
| RD | Met Office |
| JH | Met Office |
| IM | Met Office |
| ND | Met Office |
| RW | Met Office |
| JA | Met Office |
| EB | Met Office |
| NJ | Met Office |
| CJ | Met Office |
| SH | Met Office |
| PB | Met Office |

*Climate Stories* took the form of a three day (2nd-4th May 2018) residential retreat at Dartington Hall, an estate and education centre in the South West of England that is set in parkland and surrounding

countryside. Crucially, this setting provided the opportunity for participants to work close to nature in a relaxed atmosphere, whilst also being away from their usual working environment. The first two days of the retreat consisted on a combination of structured workshops in which participants experienced each of the four art forms. These events were collaborative in nature and were designed to introduce participants to different ways of conceptualising climate change and to the methods adopted within the

arts. Crucially, there were aspects of activities that were also individual, providing necessary time and space for reflection on the learning experience. The final day of the workshop provided an opportunity



for participants to select an art form that they wished to pursue in order to develop a piece of work on a chosen area related to climate change.

The evaluation of *Climate Stories,* on which this paper is based, was undertaken by one physical geographer and one human geographer with interests in climate science communication. The project enabled us to undertake a series of qualitative data collection exercises through participant reflective diaries and interviews with participants during the *Climate Stories* workshops. Through these data, we sought to explore the learning journeys and experiences of individual project participants to understand the ways in which climate scientists engaged with a range of art-science collaborations. In this way, we aimed to explore the extent to which art-science collaborations are capable of challenging scientific orthodoxies to promote sustained changes in the way in which climate scientists practice climate change communication.

Prior to commencement of the retreat, all participants provided written consent and the project received ethical approval. Participants were also guided through both the nature of critical self-reflection and ways in which they could document their feelings, emotions and learning experiences throughout their time at Dartington. To do this, participants were asked to keep a diary for the duration of *Climate Stories* in order to capture their reflections in the form of text, drawings and artefacts. In addition, semi-structured interviews were conducted on the final day of retreat and participants used the reflections in their diaries as a prompt for the interview discussions. All interviews were recorded using a voice recorder and following the project, both diary contents and interviews were transcribed. The analysis used an interpretative approach and involved a two stage coding process. Initially, open coding was deployed on all data to systematically analyse and categories emergent narratives (Mills et al., 2006), followed by axial coding as a means of relating data to uncover sub-categories within participant data (Allen, 2017).

The following sections convey three arguments. First, we demonstrate how the collaborative and supportive atmosphere at Dartington led to participants experiencing greater personal and professional confidence. Second, we explore how a series of art workshops helped participants to understand and reflect on new ways of seeing and understanding climate change. Through these activities, a strong



sense of collaborative learning revealed the importance of shared ideas and experiences. Third, we

illustrate how Climate Stories led participants to critically reflect on their standard practices of science

communication and facilitated an enthusiasm to make future engagements with publics more interesting

through mobilising different forms of art.

## 6.    Bringing the self into science

Climate scientists typically receive their training within the physical sciences, and are often employed

in institutional environments that are dominated by those of similar disciplinary backgrounds.

Accordingly, the ontological and epistemological positions of climate scientists are largely formed by

their adherence to the scientific principles and practices that dominate their daily work. Yet working

effectively outside of a scientific context requires scientists to stray from their normal practices and to

engage with new ways of seeing and knowing about the world. Whilst many participants acknowledged

past or current familiarity with the arts, we demonstrate how participation in these workshops helped to

engender a sense of liberation from routine scientific practice which promoted not only enjoyment, but

more importantly, a sense of increased personal and professional confidence. This discourse charts the

journey that individual participants took throughout the workshops and illustrates how increased

confidence emerged from their experiences. To do this, we firstly explore the initial reflections offered

by some participants. Importantly, these dairy extracts highlight a sense of apprehension representative

of perceptions and practices that prevail within a scientific working environment:

> "This is an intimidating group of highly qualified inspirational people. I hope I am able to
> apply my forecasting background effectively. It's been a while since I was in climate, they have
> taken the gamble and allowed me this opportunity. Now I need to: deliver; not disappoint, be

> engaged, be present." (IM).

> "Very out of my comfort zone. Was expecting something more like creating a play. Instead,
> less structured. Linking place and environment to ideas about research. Felt more nervous than
> usual volunteering ideas, as no confidence in their quality. Used to needing to be right in order
> for an intervention to be valid, but different for creative pursuits" (ND).



Whilst these examples are illustrative of particular concerns, many participants initially recorded a
general apprehension about working in a new environment, twinned with an excitement and sense of
challenge presented by the opportunity to participate in the project. Crucially, both diary extracts and
interviews with participants chart a growing sense of community throughout the workshop, alongside a
sense of collective endeavour to make a positive contribution to engaging publics with climate science.

At one level, this allowed many participants to feel more liberated and comfortable in exploring both
their own ideas, and in contributing to group activities. In addition to this, many participants reflected
on the strength of shared learning and emotion that emanated from the workshop activities and through
working with other climate scientists and artist researchers:

"I've found it very challenging and liberating. Because it's been such a safe space; everybody
here has come expecting to try new things, which they're very much not experts in, a feeling
that everyone's a beginner, […] a real freedom to fail" (HL).

"Today's evening entertainment was moving. The poems especially stirred my emotions and
made me want to begin a new poem of my own. It hasn't come to me yet though" (JA).

"It was amazing, very inspiring, very moving to be able to connect with your peers in this way.
We had some really magical moments when we really shared something, and we were all quite
emotionally touched" (PB).

We use these narratives to illustrate the importance of environmental setting in fostering a safe, friendly
and encouraging atmosphere in which participants could build a supportive community for learning.
Moreover, these narratives are illustrative of ways in which shared learning and experiences can
engender personal emotion and a shared sense of passion for climate change as a significant societal
challenge. In this way, many participants reflected on the happiness of working with peers and the
confidence that grew through these interactions. The following diary extracts demonstrate three
important influences of the workshop experience on the confidence of individual participants. Firstly,
there was a strong sense among many participants of the importance of collaborating in a quiet, relaxed
setting away from a normal working environment. Indeed, the strength of this approach is illustrated by



the sudden change of mood experienced by one participant when the workshop was criticised on Twitter:

> "I was walking back to the hall of residence, still feeling in a happy bubble when someone stuck a pin in – burst, happy and content feeling gone, replaced by sadness, fear, anger. Some people on twitter obviously did not like what we were doing or what this workshop was about. An hour of tweeting followed, supported by others from the workshops, and others on twitter" (RD).

Secondly, the strong sense of support between participants emerges frequently in both the diary extracts and interviews of participants. In this way, there was a clear effect of confidence-building and the formation of friendships through such collaboration:

> "Some people are out of their comfort zone and quite obviously uncomfortable…people have noticed that and been sensitive to that …and been encouraging each other in a very non-threatening and non-confrontational way. It's been lovely to see that. I think the friendships that have been formed at Dartington will last" (JA).

Thirdly, there were a number of very personal achievements noted in the diaries of participants which highlighted the long-lasting benefits of the workshop experience on increasing personal and professional confidence:

> "As we approach the end of this stage of the climate stories journey, I wanted to articulate the profound impact this has had both personally and professionally. I started this project with dyslexia and while this is obviously still the case, I have now read aloud for the first time since school […]. Who would have also thought I would volunteer for a creative writing workshop!" (IM)

Overall, these examples are illustrative of the increased personal and professional confidence that climate scientists may experience from working outside of their routine environment. Participants embraced the challenge of working in a new and potentially daunting environment, yet the physical setting and sense of collective identity created an atmosphere conducive to confidence building.





## 7. Conveying through creativity

A number of fundamental challenges may exist when artist researchers and climate scientists engage in collaboration. From a scientists' perspective, there may be concerns about how the tightly constrained practices and formalised representations of science may translate and be conveyed through art.

Moreover, there may manifest very personal concerns around the degree to which such collaborations and resulting artworks will be perceived as advocacy, and as such, how these may impact upon both the individual and the organisation they represent. Importantly, we demonstrate that these common assumptions were not realised among most of the participants. Conversely, the data reveal that the workshops served as a source of inspiration for participants, and an opportunity for effective critical

self-analysis of their scientific work in relation to different art forms. Foremost among the reflections was the enjoyment that participants experienced in understanding the opportunities afforded by different art forms (print-making, theatre and performance, creative writing and song-writing) for thinking about and engaging people with a threat perceived by many to be distant and unimportant:

"Great insights from Dan as to why climate change hasn't inspired much great art in the UK. It

needs to inspire love or anger about something; clearly about our immediate lives. Something visual" (ND).

"I really enjoyed this activity (theatre and performance workshop), because it made the link with the natural world around us, but also how it made you think about things in a completely different way – of what does this scenery, place, smell, etc. mean to me, and what could it mean

/ how could it represent aspects of my research" (GT).

"Imagining the here and now, but differently, through our individual experiences brought cloud condensation, tree ecosystems large and small, root systems and subsoil, tropical rainforests and future landscapes under climate change into view – unearthing the inviable, trying to feel what's remote or not here yet" (RD).

Within this setting, participants engaged with each art form and consequently reflected on their experiences of learning. As such, participants were able to find art forms that gave them a sense of both





enjoyment and challenge, alongside an opportunity to further develop their ideas for communicating

climate change. Below, we present a series of narratives that illustrate the differing experiences of three

participants in one of the activities (the print-making workshops). We present these narratives to

illustrate the process of critical self-reflection that participants engaged in during their stay at

Dartington:

> "The exercise overall is a bit self-promotional for me, but I think that as a scientist I need to
> become better at promoting my work. So, the exercise has perhaps made me slightly more
> comfortable with doing this" (JH).

> "I found I lacked the patience and I also found the concentrated quietness of everyone not to
> my liking, in the end opting to use my iPhone to supply music in my ear pieces. The inking in
> was also much harder than I initially thought and I struggled to get good results. I think my
> design was too complicated for my lack of patience" (RD).

> "My activity of choice on the final day was print making. Our task was quite structured, with a
> 'talking heads' theme. Fiona taught us new skills and was very generous with her time,
> materials and guidance. If afforded the opportunity to do this again, I find myself now with a
> collection of climate visualizations ideas I would like to explore further" (IM).

These examples highlight the value of participants taking the time to engage with other climate scientists

and artist researchers to both imagine how their climate knowledge could be conveyed through forms

of art and to explore their personal preferences for different ways of working. Importantly, participants

reflected on the importance of having time to engage in collaborative group activities and discussion,

one-to-one conversations, and individual reflection since all provided different opportunities for

learning. For example, participants commented on the importance of having time to develop their ideas

with artist researchers, as well as the space to reflect and work on their project individually. In addition,

many participants highlighted the ways in which group work provided a very constructive and

supportive environment for sharing very different perspectives and ideas, whilst ensuring that



knowledge and ideas were valued on an equal basis. The following extracts, alongside Figure 1, describe some of the key benefits of collective learning noted by the participants:

"The group work has been great, because, using all of those different experiences you get so many different ways of looking at things. Some of our creations have been solo…and some of them, like the song writing, have come out of us blending our ideas together" (FB).

"Getting into in-depth conversations about how we see and perceive the world…everyone brought something that enriched the group's experience" (RD).

"I love that you can get five people and give them the same task and get 5 completely different outcomes!" (JA)





**Figure 1** – Totem Banner – A collective art (Tread Lightly on the Earth), created by participants at the

first print-making workshop (Dartington). Photograph: P. Thomet.





> "So, the banner was from the deep sea, to the coast, the shore, forest, and going up to the sky. It was really funny because we all had our own specific interest and we were all keen to have an input into our favourite area. So mine, I've always been obsessed with clouds, I work on monsoon and rain so I just wanted to do the top bit [laughter]. So people started at the bottom and they did their corals and things and they started with my clouds and, helping each other at the same time so we are not completed isolated. We started at both ends and we met in the middle and it was, yeah, it was fantastic" (PB)!


Overall, these findings reveal three important outcomes relating to art-science collaborations within this

setting. First, the participant reflections illustrate a willingness and enthusiasm to explore other (non-scientific) ways of seeing and coming to know about climate change. In so doing, there was a widespread recognition of the importance of different art forms as ways of making climate science both personal and potentially more relatable to wider audiences. Second, the ability of participants to engage in effective critical self-reflection illustrated the importance of having time and space during the

workshops to create an immersive experience in which individuals can find an art form and conceptual focus which they feel comfortable in pursuing. Thirdly, in addition to building personal and professional confidence, there was a clear sense of the academic value of collaborative activities and discussion in promoting effective sharing of ideas in an environment devoid of knowledge hierarchies. Whilst acknowledging that the effectiveness of these outcomes was contingent upon many factors, including

group outlook, dynamic and environmental setting, these results nonetheless provide evidence that successful art-climate science collaborations may be achieved over a short period of time.

### 8. Sustaining *storytelling* in climate science practice

One of the most significant questions relating to art-science collaborations is the extent of their influence on the professional practice of the participants. Are such interactions short-term meetings of minds that

are very much of the moment, or is there evidence for more medium to long-term impacts in the form of sustained interest in art-science collaborations and shifts in professional scientific practice? This final theme emerged from interviews with participants that took place on the final day of the Dartington



workshop. Crucially, these reflections reveal the ways in which participants were able to critique their standard working practices and explain their intentions to review their approaches to climate science

communication. This culminated in a collective anthology of art works which represented the individual and collective efforts of the project participants and illustrated the potential of climate storytelling as a means of communicating science.

As we have argued throughout this paper, the deficit approach remains a dominant mode of communication within climate science. This extract illustrates how one participant reflected upon their

routine communication practice recognising the flaws inherent in the deficit model:

"I think the challenges are…that I'm aware that I've been in a broadcast mode, and I have typically seen communication as 'I have knowledge and I am wanting to communicate it to people'. Hey, this is this really exciting fact that I found out about our weather, you all want to know about this – it's great. And some of the challenges I think are that there's so much

information content, particularly nowadays is so large and so out there, that people […] I wonder now if the challenge is that people are overwhelmed by the amount of information that we feed them, and that perhaps exploring different ways, like we are here, is useful to see well maybe there are other ways to engage and make that outreach and link to people" (SH).

Through the creative, communal and supportive atmosphere formed at Dartington, there was a clear

sense of personal and collective emotion associated with the climate change experiences relived and shared by participants. Whilst we chart the impact of this on the confidence of individuals in section six, importantly, participants recognised the role of conveying and inspiring emotion through storytelling for engaging publics with climate change. The following extracts illustrate the ways in which participants intended to develop their art works to transform their climate change communication

and bring emotion into the dialogue:

"By using art and the emotions that art elicits within us, we can maybe really start to reach people who haven't thought about these issues before, and get them thinking about things in



new ways and really considering the impact of climate change on the world around us and thinking about how it really is going to affect our lives in the future" (FB).

"This idea that climate change is difficult to express artistically, or perceived to be, and you know, it doesn't often come up in the charts and songs and you know, it's often seen as a bit of a boring topic, I guess, because there's no emotion attached to it, basically, there isn't traditionally strong feelings attached to it. Whereas, I think that's something I would really like to try and talk about and work with people towards because that's the polar opposite to my
experience of it. When you're snorkelling around on the Barrier Reef, or when you're sailing through the arctic, and you're seeing just coral rubble-fields and ice melting into the sea, it's heart-breaking, it's really, very, very emotionally strong. So to see it become a topic that's dry and emotionless, it's not right, it's a wasted opportunity. We're talking about it in the wrong way! So, all of the workshops here explore ways in which we can bring emotion into the
dialogue, but I think creative writing is definitely one of those" (HL).

These examples highlight the strong desire from participants to make a tangible difference to the ways in which climate science communication is undertaken. Crucially, this transformatory behaviour led to the production of a collective publication (*Climate Stories: we all have a story to tell about climate change* – available online), which outlines the ways in which the participants enthusiastically engaged
with different art forms and went on to create multiple pieces of art with the aim of enhancing the engagement of publics with climate change. In addition, participants reflected in interviews on the ways in which their experiences at Dartington had changed their perception of science communication and importantly, how it had made them review their normal working practices:

          "I would say for me, the main take-away has been the opportunity to take a step back and be
595        pushed into looking at what I do from quite a different perspective. Being given some techniques and methods for adopting a different mindset. I think it's very difficult sat in your normal space, at my normal desk to try and do that. So, being in a different environment, being with different people, and being posed different questions that I wouldn't think to ask myself prompt me to step back and re-evaluate how I think about what I do" (SH).





"I found that it (the workshops) really helped me to change my perspective, and have a much

clearer message, to try and simplify and make it more striking, personal, relevant to people,

rather than facts, numbers and evidence. So that will definitely stay with me and I've been

thinking about how to include that in my science communications much more" (PB).

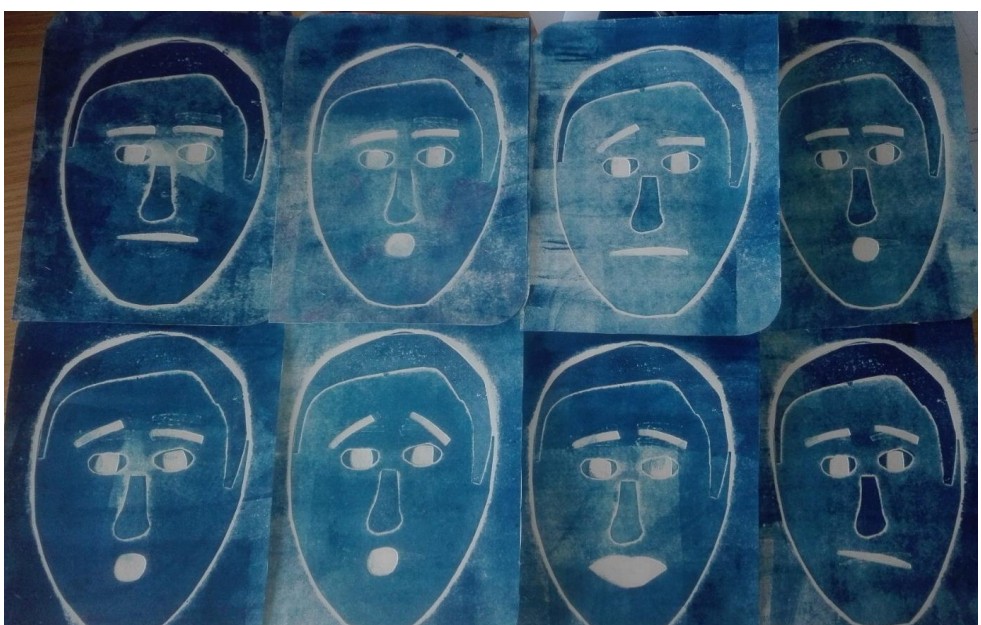


Figure 2 – A print produced by one participant. The dairy caption underneath read "~~Scientist?~~

~~Networker? Coordinator?~~ Communicator". This example is illustrative of the ways in which some

participants used different art forms to reflect on their professional roles (JA). Photograph: J. Gaunt.


Overall, these narratives illustrate the significant impact that this art-science collaboration had on the

ways in which individual participants viewed their standard practices to science communication. As

such, the extracts demonstrate not only a recognition of the ineffective nature of deficit communication,



but also the enthusiasm for experimenting with new ways of engaging publics through storytelling. The

importance placed by participants on the role of emotion, and their willingness to contribute their

artworks in a publically-accessible manner, illustrate the comparative comfort in engaging with

advocacy at a level not usually adopted within climate science. Arguably, the most significant outcome

was the desire expressed by some participants to sustain their critical reflection on communication

practices and to embed their new understanding within future science communications and engagements

with publics.

### 9. Discussion

Throughout this paper, we have argued that the climate science community must evolve its practices of

science communication and engagement with publics in order to address fundamental changes in the

relationships between science and society. Accordingly, we argue that climate scientists not only need

to move beyond the predominant use of deficit model communications (Illingworth et al., 2018), but

those seeking to engage in arts-based climate communication need to critically evaluate the potential

limitations of employing scientific framings of advocacy (Donner, 2014; Schmidt, 2015; Schmidt and

Donner, 2017) in their own practice. In addressing both the need for climate scientists to explore the

issue of climate advocacy, alongside new and exciting ways of engaging publics with climate change,

we have argued that the arts provide an exciting opportunity for addressing current communication

challenges (Nurmis, 2016; Galafassi et al., 2018). We suggest that climate scientist-artist researcher

collaborations may provide social learning opportunities for climate scientists in order to transform their

science communication practices. In making this argument, we seek to make three contributions to

research and scholarship on climate science communication, climate science practice, and art-science

collaborations.

First, the evidence presented suggests that science-art collaborations within specific contexts can lead

to increases in the personal and professional confidence of climate scientists. Importantly, whilst some

climate scientists demonstrated an initial discomfort in working outside of their routine practices, there

was a widespread acknowledgement of the limitations of positivist disciplines in engaging with values,

purpose and meaning (Hulme, 2011). As such, researchers were very open to discussing their personal





emotional responses to climate change, despite the paucity of such discussion within the western cultural context of scientific practice (Head and Harada, 2017). Emerging from our research is a clear sense of the importance of creating appropriate environments that are conducive to effective art-science collaboration. Indeed, the potential of residential art-science retreats situated in remote natural environments has been highlighted in the literature as an effective means of stimulating informal, non-judgemental discussions about climate change (Jacobson et al., 2016). However, we argue that more localised, green environments (formal gardens, countryside) provide an atmosphere equally conducive to effective learning through access to nature for inspiration, reflection and relaxation, as well as a geographical disconnect from a routine work environment. Crucially, our findings demonstrate the positive influence on climate scientists of collaborative learning within such environments. In alignment with other findings, we demonstrate how a strong sense of community among climate scientists can be borne out of working towards a shared goal, a process that can provide both empowerment and meaning (Clayton, 2018). In addition, we show how engagement with the arts provides the potential for bringing out emotion in scientists, and even creating a celebratory atmosphere of their work (Curtis et al., 2012). As such, we argue that working collectively can lead to the development of new social relationships, important sources of social support and increases in self-esteem (Clayton, 2018; Bamberg et al., 2018). Crucially, our findings recognise the importance of understanding the role of emotion on climate change and how this goes beyond current rational and scientific practice (Head and Harada, 2017).

Second, we argue that collaborative art-science learning can enable scientists to engage effectively with new ways of seeing, knowing about, and expressing climate change and its impacts. The principal challenges of engaging people with climate change relate to its slow evolution, its distance in both time and space, and its often abstract and socially distant nature (Stoknes, 2015). Here, we demonstrate that through engaging with different art forms (print-making, creative writing, theatre and performance, and song-writing), climate scientists can seek to overcome these barriers by moving outside of the working constraints of scientific orthodoxy. Importantly, our findings support the notion that the arts can encourage climate scientists to invoke their individual and collective imagination, one of the most important concepts in establishing a human relationship with climate (Nurmis, 2016). As such, we find



that collaborations can create spaces in which active experimentation and imagination are capable of

encouraging creative thinking (Kagan, 2010), a finding that emerges repeatedly in workshop reflections

of participants and in their artworks. In this way, artistic practices permit freedoms to engage with

multiple realities that can effectively connect climate change to many other human challenges

(Galafassi et al., 2018). The research also revealed advantages that can stem from working in a

collaborative art-science environment. We suggest that in addition to providing opportunities for

transforming practice, such participatory spaces can lead to shared and negotiated understandings of

existing knowledge (Gibbs, 2014; Paterson et al., 2020), a key aspect of non-hierarchical learning. In

addition, such activities place an emphasis on social interaction and by their nature provide support for

participants. Cumulatively, these processes are conducive to effective social learning on new ways of

communicating climate change to publics.

Third, our project demonstrates the potential for embedding and sustaining climate storytelling within

scientific practice in an effort to engage a more diverse range of publics with climate change and its

impacts. Importantly, our research revealed that by the end of the *Climate Stories* workshops, many

scientists were able to reflect critically on their standard communication practices and recognise the

complexities and deficiencies inherent within the deficit model (Simis et al., 2016). We demonstrate

that through engaging with different art forms, scientists identified the possibilities for developing

engaging narratives to communicate their research, despite the negative connotations of storytelling that

commonly occur within the scientific community (Dahlstrom, 2014). Indeed, our findings support the

notion that storytelling can provide insight into ways of improving the effectiveness of climate change

communication (Martinez-Conde and Macknik, 2017). Alongside this, the artwork produced on *Climate*

*Stories* illustrates the wide range of opportunities for representing within stories climate change

characteristics operating at different geographical scales (Daniels and Endfield, 2009). Crucially,

research has indicated that narratives framed as stories have the potential to outperform factual climate

narratives for encouraging action on climate change; potentially a result of the former eliciting greater

autonomic reactivity and emotional arousal (Morris et al., 2019). Accordingly, we demonstrate how art-

science collaborations not only hold the potential for engaging climate scientists with new ways of



seeing and representing their work, but also provide a basis for these individuals to develop their ideas further and create sustained interventions in their routine communication and engagement practices. Nonetheless, we note that climate scientists must enter the process of storytelling with an understanding of the paradox associated with this style of communication:

700   "…how can science preserve its credibility as curator of knowledge while engaging audiences with a communication format that is agnostic to truth?" (Dahlstrom and Scheufele, 2018: 1)

In addressing this complex issue, we argue that it is necessary for scientific institutions to re-evaluate the support that they provide to scientists wishing to engage in art-based science communication and engagement on climate change. We recognise that art-science collaborations are most likely to be self-

705   selective and will appeal to those with genuine interest, past experiences or double qualifications (Rödder, 2017). Nonetheless, we suggest that in order for these promising developments to be sustained, the climate science community need to re-evaluate the knowledge hierarchies and epistemological constraints that hinder advances in science communication. Alongside this, there is a requirement for funding bodies and scientific institutions to recognise the significant value of collaboration with the arts

710   and humanities to enable scientists to become more comfortable and effective climate change communicators.

### 10. Conclusion

Recent years have witnessed science operating within a transformed societal context marked by an erosion of trust in the scientific enterprise and a diminished social status of scientific knowledge. Whilst

715   climate scientists have endeavoured to keep pace with these changes, effective science communication needs to move beyond an over-reliance on the deployment of large-scale deficit-style communications, alongside a common adherence to assumptions around the objectivity and neutrality of scientific practice. In order to address these challenges and provide a greater opportunity to engage diverse audiences with climate change, we advocate that climate scientists consider innovative and creative

720   ways to communicate with publics through different art forms, whilst simultaneously seeking to develop conceptual understandings of advocacy that go beyond scientific frameworks. We demonstrate that



through collaborative engagement with a range of artistic practices and disciplines, climate scientists may be afforded opportunities for re-imagining climate change in ways that transcend scientific practice.

Through this research, we have demonstrated that collaborative art-science learning is capable of engendering a heightened sense of personal and professional confidence through providing a learning environment conducive to shared ideas and goals in a non-hierarchical environment. In this way, collective learning about climate change through the arts is capable of invoking cultural and emotional responses that are absent in most professional scientific discourses. We highlight that collaborative art-

science collaborations can provide the setting for climate scientists to reflect critically on the ways in which art forms can be pursued to develop novel climate stories with which to engage publics. In particular, we show how collaborative art-science learning encourages climate scientists to engage in discussing ideas and creating negotiated (shared) understandings of how science may be represented through art forms. From this process, we show how art-science collaborations of this nature are capable

of allowing climate scientists to learn about and become comfortable with their personal position on climate advocacy. Equally important is our assertion that these types of activities can equip climate scientists with the skills, networks and enthusiasm for sustaining arts-based interventions within their climate communications practices. Nonetheless, we recognise that pursuing these developments will require a number of transitions within the scientific community. First, the climate science community

must recognise the weaknesses in current communication practices and the opportunities afforded through working with the arts. Second, greater recognition of the role and importance of art-science collaborations for engaging publics with climate change must be recognised by research councils and funding bodies to support this area of academic work and outreach. Third, scientific institutions must recognise the role and importance of art-science collaborations through re-evaluating how they

professionally value and support contributions made by scientists in this area. Fourth, we call for much greater recognition of the potential for collaborations between the climate sciences and the arts and humanities through transdisciplinary projects. In calling for these transitions, we seek not only to argue for the role of science-arts collaborations as a means of more meaningfully engaging publics, but also



to re-frame the role of scientists to recognise the vital role they might play in telling their climate stories

through emotionally-connected and engaging practices.

**Author contribution**

EW prepared the manuscript with contributions from co-authors. EW and SB designed and conducted

the project analysis, supported by CRo, RP and RS. PS, PT and CR led the project design. SF, FL,

EO'M, and DP were artistic project leads.

**Competing interests**

The authors declare that they have no conflict of interest.

**Acknowledgements**

The authors would like to thank all those who participated in Climate Stories and the Natural

Environment Research Council for financial support to undertake the research, grant number:

Funding: This work was supported by the Natural Environment Research Council: NE/R011729/1.

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
