# Peer review of "Climate Stories: Enabling and sustaining arts interventions in climate science communication"

_Geoscience Communication, 2022_

## Referee Comment (RC2)

**Review for GC**

**General Comments: GC-2022-7**

Thank you for the opportunity to review this interesting and timely article: Climate Stories: Enabling and sustaining arts interventions in climate science communication. This is an important and valuable paper addressing relevant scientific questions that are well within the scope of Geoscience Communication, and I believe that it will be of keen interest to readers of GC. The paper clearly outlines some of the challenges that face the climate science community in communicating environmental risks and offers a very comprehensive critical review of existing dominant deficit approaches to communicating climate change.

My main reservation with the manuscript as it currently stands is that a couple of sentences in the abstract and opening sections, related to claims around engaging wider publics, appear to 'over promise' (details and specific examples are presented in the review below). However, this point might simply require a minor revision to ensure consistency with the stated aims throughout the paper. Otherwise the results presented throughout this paper are sufficient to support the interpretations and conclusions. Other minor suggestions to strengthen this paper, including potential to reduce some repetition in the sections 2 and 3, are detailed in the review. In summary, it is a very engaging and accessible paper, detailing an innovative collaboration, and I highly recommend it for publication in this journal.

**Specific Comments:**

Below I have included a number of specific comments which I hope will serve to strengthen the manuscript.

- The opening sections seem rather repetitious and I believe that might be because of the length of Sections 2 and 3 which could be reduced.

- The evidence presented in the case of the collaborative workshop currently does not support one of the stated aims highlighted in abstract and the opening sections of the paper e.g. this extract from the abstract states:
  *the paper highlights innovative ways in which climate change communication can be re-imagined through different art forms to enable complex concepts to become knowable, accessible and engaging to wider publics....*
  The case of collaboration which forms the central part of this paper does not currently provide enough details on any engagement with the wider publics to support this claim. I believe that this issue can be addressed by softening or rephrasing this aim within the abstract and opening section. Indeed, at Line 102 this aim is rearticulated and expanded and it is rephrased to more accurately summarise the significance of the research:
  *we demonstrate that engaging in art-science collaborations offer climate scientists opportunities for gaining increased personal and professional confidence, enhanced and widened intellectual engagement with climate change, as well as 'opportunities for creating new and potentially effective means of engaging publics with climate*

*change and its impacts'.* Please review with an eye to ensuring consistency with the stated aims throughout the paper.

- In a similar vein, at Line 687, please clarify if this claim ('our findings support the notion that storytelling can provide insight into ways of improving the effectiveness of climate change communication') is based on the wider Climate Stories project as opposed to the findings in this paper i.e. workshop, which is limited in making claims about the effectiveness of these approaches.

- Line 75 –This claim (we explore ways in which climate change may be made more emotionally connected and engaging to a diverse range of publics) might require clarification or rephrasing as it could be argued that the participants in the case study collaboration described later in this paper do not represent a diverse range of publics.

- While the title of the paper is accurate, the manuscript might benefit from a more explicit statement about the limitations of this paper i.e. it doesn't set out to test the effectiveness of this approach for wider publics, focus is on climate scientists.

Overall I enjoyed the results and discussion sections and as a geographer who wishes for more time for communication activities and improved communication strategies and skills, I found myself relating to many of the participant quotations included in this great paper - e.g. discussion around self-promotion on Line 487 – and I'm prompted to read more about the larger study.

**Technical edits/suggestions:**

Phrases like 'In so doing', and 'In addition' could be reduced throughout the text

Lines 395 and 607 – insert diary instead of dairy

---

## Author Response (AR1)

**Response to reviewer comments**

Thank you very much for the reviews of our manuscript and the detailed comments provided by the reviewers, as well as your editorial steer. We have considered the reviewer comments and your editorial recommendations, and these are detailed below.

**Editor's comments**

While your manuscript is underpinned by substantive evidence-based research, please clarify the link between your aims, methods, results and conclusions throughout the text. Illustratively, 'we argue' in the abstract appears to precede any mention of data collection.

Response: we have removed reference to our argument in the abstract prior to mentioning our empirical work. We have also addressed your wider point relating to the links between our aims and the rest of the paper by making adjustments to our introduction section, as noted in response to other reviewer comments (see below).

Additionally, it would help to distinguish the method used in the activity (ClimateStories) from the methodology for the research in this submission.

Response: we have made explicit that we used an interpretivist, qualitative methodology, with specific methods to collect our data.

**Reviewer 1: Tiziana Lanza**

I believe that the present paper is of great interest for the journal and for the climate scientists and geoscientists community. The authors intend to explore the extent to which art-science collaboration are capable of challenging scientific orthodoxies to promote sustained changes in the way in which climate scientists practice climate change communication. And overall the paper encourages such a collaboration, which is positive and welcomed.

Response: thank you for your supportive comments.

Nevertheless, I believe that the paper should be re-organized, since there are too many repetitions in the manuscript text. The paper as it is, is confusing. For instance, results and findings are enumerated along the text rather than having a specific section, that in my view would enhance the results. While I consider important the numerous references to the literature that inspired the project, I believe that the introductory paragraphs are too long compared to the description of the project itself. We know that nineteen participants gathered together for three days in a beautiful and inspiring environment. We also know that they experienced four forms of art, but nothing Is said on how the workshops were actually organized. To figure it out what the participants did during the three days' retreat, we only have a couple of examples in the two figures, that are examples of what happened during the last day. Nothing is said about why you chose these forms of art. It would be really nice to learn more about the project. Same observation for the data collection and the evaluation of the experience. We know that the evaluation is based on the diaries of the participants and that they were also interviewed. But nothing is said about the questions posed during the interviews. At the same time the process of analysis is not properly described, and it would be the case to expand on what is written in line 370-374. All this said, I consider the present study interesting and original since focused on the scientists that are at the origin of the communicative process. it would be appropriate to organize the article in the right way.

Response: we have extensively reviewed the opening paragraphs of the manuscript to ensure that the flow of material is logical. We have made several changes to enhance the flow and ensure the logic of the paper's arguments are clear.

Specific comments:

31-33 "In doing so, the paper highlights…" Can you please check where in the paper this has been accomplished?

Response: we have amended this sentence, to remove reference to direct working with publics, so as to avoid over-claiming.

33. "We demonstrate…" Very often you use this term. I would rather rephrase all over the paper in "our study suggests" since the sample used for the research is nineteen participants, from one country and two Institutions. Not a large sample to allow you to demonstrate something. Or very probably your study confirm what is stated in the literature quoted in par. 4.

Response: we have amended the text to use different phrases in some cases where these are appropriate.

74 "In addition" too many times repeated (see 85 and in the rest of the paper) Please shorten par. 2 and 3. In particular, in par. 3 do you think it is worth spending all these words on the science-advocacy continuum if you declare in 228-230: "Yet, whilst the science-advocacy continuum (Donner, 2014) may be of value for mainstream communications, we argue that it is of limited utility to climate scientists who wish to explore more radical and experimental ways of engaging people with climate science through different art forms". Consider merging paragraph 4 into the introduction once you have shortened it.

Response: this has been undertaken, with a shortened introduction section to bring prominence to the intellectual debates in sections 2 and 3.

311-314 I believe you already stated this in the former paragraphs.

Response: we have checked this for consistency and believe there is no repetition.

366 "To do this, participants were asked to keep a diary for the duration of Climate Stories". Since the diary is a form of intimistic writing it would be interesting to know if the participants knew they were participating in an experiment, and it would be worth asking whether this could have influenced the spontaneity of their writing.

Response: we have clarified that, in line with our ethical procedures, participants were aware that diary entries would be used as part of the evaluation of the project.

371-374 Please expand on this. Give examples of the analysis you performed on the transcription of the diaries and interviews. There are no traces of this analysis in the article. In the following paragraphs you report only some excerpts from the diaries of the participants and from the interviews.

Response: we have used a standard social science practice of analysing the data and reporting them in this paper, in which selected quotations are mobilised to support our arguments, based on the thematic analysis undertaken. It is no normal practice in our experience to provide further analytical content in addition to the illustrative quotations used in this paper.

Consider merge par. 6 and 7 and 8 in one par with sub paragraphs (if needed) titled results. Maybe you can add a tab. were you you can summarize everything that has been achieved like: 1) they felt comfortable exploring their own ideas and at the same time contributing to the group's activities. 2) shared learning and experiences engendered personal emotion and a shared sense of passion for

climate change and so on… Finally shorten discussion and conclusion, since are repetitive of what has been already said.

Response: we would like to maintain the current structure, given that we have discrete arguments to pursue, aligned with the empirical evidence. We feel that the paper's current empirical structure makes it easy for readers to navigate and to discern our points. We prefer not to shorten the discussion and conclusion, as these sections are not repetitious and convey our arguments.

**Reviewer 2: Frances Fahy**

This is an important and valuable paper addressing relevant scientific questions that are well within the scope of Geoscience Communication, and I believe that it will be of keen interest to readers of GC. The paper clearly outlines some of the challenges that face the climate science community in communicating environmental risks and offers a very comprehensive critical review of existing dominant deficit approaches to communicating climate change.

Response: thank you for your very supportive comments.

My main reservation with the manuscript as it currently stands is that a couple of sentences in the abstract and opening sections, related to claims around engaging wider publics, appear to 'over promise' (details and specific examples are presented in the full review). However, this point might simply require a minor revision to ensure consistency with the stated aims throughout the paper. Otherwise the results presented throughout this paper are sufficient to support the interpretations and conclusions.

Response: we have re-drafted the sentences related to some over-claiming, as noted by this reviewer (see below).

Other minor suggestions to strengthen this paper, including potential to reduce some repetition in the sections 2 and 3, are detailed in the review. In summary, it is a very engaging and accessible paper, detailing an innovative collaboration, and I highly recommend it for publication in this journal.

Response: thank you for these positive comments.

Specific Comments: Below I have included a number of specific comments which I hope will serve to strengthen the manuscript.

• The opening sections seem rather repetitious and I believe that might be because of the length of Sections 2 and 3 which could be reduced.

Response: we have amended the introduction to the paper, which did contain material that was repeated in sections 2 and 3. This has liberated sections 2 and 3 to be the main intellectual basis for the paper, and shortened the introduction to make it succinct.

• The evidence presented in the case of the collaborative workshop currently does not support one of the stated aims highlighted in abstract and the opening sections of the paper e.g. this extract from the abstract states: the paper highlights innovative ways in which climate change communication can be re-imagined through different art forms to enable complex concepts to become knowable, accessible and engaging to wider publics.... The case of collaboration which forms the central part of this paper does not currently provide enough details on any engagement with the wider publics to support this claim. I believe that this issue can be addressed by softening or rephrasing this aim within the abstract and opening section. Indeed, at Line 102 this aim is rearticulated and expanded and it is rephrased to

more accurately summarise the significance of the research: we demonstrate that engaging in art-science collaborations offer climate scientists opportunities for gaining increased personal and professional confidence, enhanced and widened intellectual engagement with climate change, as well as 'opportunities for creating new and potentially effective means of engaging publics with climate change and its impacts'. Please review with an eye to ensuring consistency with the stated aims throughout the paper.

Response: we have amended the abstract and the final sentence of the introduction to ensure the paper's claims are in line with our evidence.

• In a similar vein, at Line 687, please clarify if this claim ('our findings support the notion that storytelling can provide insight into ways of improving the effectiveness of climate change communication') is based on the wider Climate Stories project as opposed to the findings in this paper i.e. workshop, which is limited in making claims about the effectiveness of these approaches.

Response: we have amended this sentence to ensure it is clear that our findings suggest storytelling may be a constructive means of enabling publics to engage with climate change.

• Line 75 –This claim (we explore ways in which climate change may be made more emotionally connected and engaging to a diverse range of publics) might require clarification or rephrasing as it could be argued that the participants in the case study collaboration described later in this paper do not represent a diverse range of publics.

Response: we have amended this sentence to make clear that storytelling has the potential to be used as a mode of science engagement in general terms.

• While the title of the paper is accurate, the manuscript might benefit from a more explicit statement about the limitations of this paper i.e. it doesn't set out to test the effectiveness of this approach for wider publics, focus is on climate scientists. Overall I enjoyed the results and discussion sections and as a geographer who wishes for more time for communication activities and improved communication strategies and skills, I found myself relating to many of the participant quotations included in this great paper - e.g. discussion around self-promotion on Line 487 – and I'm prompted to read more about the larger study.

Response: we have amended the end of the conclusion to make this clear.

Technical edits/suggestions: Phrases like 'In so doing', and 'In addition' could be reduced throughout the text

Response: we have edited the text accordingly.

Lines 395 and 607 – insert diary instead of dairy

Response: completed.

---

## Author Response (AR2)

**Response to the Editor (8th August 2022)**

Dear Dr Arnal,

Many thanks for informing us of your acceptance of our manuscript, subject to final review of the minor corrections that you have requested. Thank you for your time in making these comments. We have addressed each comment, and these are outlined below and reflected in the amended manuscript.

Best wishes,

Ewan Woodley

**Editor comments:**

- Echoing initial comments from Referee #1, please consider providing a sample of the questions asked to the participants during the semi-structured interviews.

**Author corrections:**

Thank you. We have inserted the questions that we used in the interviews with participants. These were prompts for an extended discussion. We have also adjusted the text to contextualise the addition of Table 3 (the interview questions).

**Editor comments:**

- Please clarify how your project was "built on" WAMFest (P12). Did these happen around the same time, with some of the same artists, and similar events/workshops?

**Author corrections:**

Thank you. We have added in dates of two key WAMfest events in 2012 and 2016 (alongside their location). We have also confirmed that some of these collaborations continued into Climate Stories, whilst the project also engaged with new artistic partners and research participants.

**Editor comments:**

- As per initial comments from Referee #1 regarding the structure of your results sections, please clearly refer to the sections that address the points you raise in the last paragraph of Section 5. Simply adding the section number and/or title in parentheses behind each point should be sufficient.

**Author corrections:**

Thank you. We have added the section numbers in parentheses to clarify the results sections to which we are referring in the final paragraph of Section 5.

**Editor comments:**

- Please consider anonymising the artists (e.g., "Dan" and "Fiona" on P19-20), as was done with the climate scientists in Table 1.

**Author corrections:**

Thank you. We have anonymised these two artistic project leads and we have inserted details of these individuals in a new 'Table 2', to provide necessary context to the reader.

**Editor comments:**

- Does the box have a specific meaning on P23?

**Author corrections:**

Thank you. The box (with text inside) on P23 is designed to sit below Figure 1; however, we have removed the box, as it does not affect the meaning of the text.

**Editor comments:**

- If possible, please add a link to the collective publication you mention on P25 L594.

**Author corrections:**

Thank you. We have included a link to this publication in the reference list. We have inserted additional text to clarify this to the reader.